# A Weakly-Supervised Method for Jointly Embedding Concepts, Phrases, and Words

## Abstract

Recent work on embedding ontology concepts has relied on either expensive manual annotation or automated concept tagging methods that ignore the textual contexts around concepts. We propose a novel method for jointly learning concept, phrase, and word embeddings from an unlabeled text corpus, by using the representative phrases for ontology concepts as distant supervision. We learn embeddings for medical concepts in the Unified Medical Language System and general-domain concepts in YAGO, using a variety of corpora. Our embeddings show performance competitive with existing methods on concept similarity and relatedness tasks, while requiring no human corpus annotation and demonstrating more than 3x coverage in the vocabulary size.

## 1 Introduction

Recent vector space models of semantics are primarily built on words. Word embeddings are either used directly as features in downstream modeling tasks (Chiu et al., 2016b), or are composed to produce representations of more complex linguistic units, such as phrases, sentences (Socher et al., 2011; Le and Mikolov, 2014), or even documents (Yang et al., 2016). However, there are many cases in which words are not atomic semantic units, and multi-word expressions cannot be expressed as a composition of their member words: for example, "the Big Apple" is not a fruit, and "Lou Gehrig's disease" has little to do with baseball. Furthermore, such concepts often multiple textual forms: "the Big Apple" and "New York City" both refer to the same location, and "Lou Gehrig's disease" and "amyotrophic lateral scle-

rosis" are the same medical condition. Despite the lack of lexical overlap between these phrases, we would like a semantic model that can represent the underlying concept, regardless of the specific textual form used.

Our novel approach to this task combines structured knowledge with proven techniques for learning word embeddings to train context-based representations for contexts, phrases, and words. The model requires no human annotations: we use known phrases as distant supervision in distributional similarity training over an unannotated corpus. As in the formulation of Mintz et al. (2009) in extracting textual relationships, we assume that any occurrence of a known phrase signifies an occurrence of each concept[1] it can refer to, and train our concept embeddings using the contexts of their phrase forms. We experimentally validate our approach by learning embeddings for biomedical concepts and real-world entities. An evaluation on concept similarity and relatedness tasks shows that our embeddings are competitive with prior models that required human annotations for concepts. We also present a novel dataset of similarity and relatedness of real-world entities, identified both by Wikipedia page and text phrase.

Furthermore, analysis suggests the source ontology structure is reflected in the organization of our embeddings: concepts and their representative phrases are embedded close to one another, and embedded concepts of the same semantic type cluster together with some regularity.

## 2 Related Work

Single word embeddings have been directly used as entity/concept models in prior work, includ-

---

[1]The term "concept" has been used in prior literature to mean both a specific concept in an ontology and a specific sense of a word. In this study, "concept" refers to a canonical abstract concept or real-world entity.

ing in analogy completion tasks (Mikolov et al., 2013b; Linzen, 2016; Gladkova et al., 2016) and in similarity/relatedness tasks (Muneeb et al., 2015; Chiu et al., 2016a). Multi-word concepts have also been explored by approaches such as lexicalization and word averaging (Mikolov et al., 2013a; Socher et al., 2013). Fan et al. (2015), Yu et al. (2016), and Hill and Korhonen (2014) modify embedding training with additional information, such as named entity mentions and associated perceptions.

Additionally, a number of models have been proposed for the related task of separately embedding different word senses. Camacho-Collados et al. (2015) use a semantic network to train word sense representations using relevant subsets of Wikipedia. Nieta Pina and Johansson (2016) use WordNet graph structure as context for embedding word senses; a similar approach was taken by Hu et al. (2015) for embedding entities, using the hierarchy of the containing ontology augmented with textual information.

In all of these cases, however, words are used as atomic units, and can be composed in order to model concepts. However, there is also a rich literature on learning embeddings of knowledge base entities and ontology concepts directly from the structure, relationships, and attributes encoded in the knowledge base (Bordes et al., 2011, 2013; Wang et al., 2014b; Lin et al., 2015). Toutanova et al. (2015) decomposed textual relations into single words to improve generalization, but still relied on knowledge graph structure for embedding entities; Socher et al. (2013) and Yang et al. (2015), however, incorporated pre-trained word embeddings as an initialization for entity representations, using word averaging.

Several alternative methods for embedding concepts atomically have been proposed in the biomedical domain. De Vine et al. (2014) automatically tag a corpus for biomedical concepts and use the sequence of concept identifiers as training context for `word2vec`, while Mencia et al. (2016) use the text content of documents tagged with MeSH headers to learn representations for the headers. Choi et al. (2016a) in contrast, use sequences of human-annotated medical codes to learn embeddings; Choi et al. (2016b) take a similar approach, and also learn concept embeddings from concept identifier co-occurrence statistics.

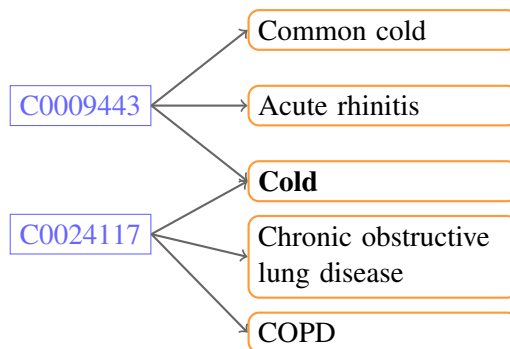

Figure 1: An example of mapping from concepts to phrases in the UMLS, and the set of unique phrases used in the mapping. The common phrase *cold* has been marked in bold.

Wang et al. (2014a) is the closest approach to ours, as they propose methods for jointly embedding KB entities and words into the same space. However, their embedding method relies on the binary relations between entities in the knowledge base, which have been shown to be highly incomplete (Min et al., 2013). Their best results also rely on human annotation via Wikipedia links.

## 3 Method

We propose a method to jointly embed concepts, phrases, and words into a real-valued space, using structured knowledge from an ontology and an unannotated training corpus.

Let $C$ denote the set of canonical concepts in an ontology. For any specific concept $c \in C$, let $P_c$ denote the set of phrases that can be used to represent it; $P$ denotes the union of all of these sets of phrases. Figure 1 illustrates this many-to-many relationship between concepts and phrases.

Additionally, let $T$ denote the sequence of tokens in a training corpus, and $W$ be the word vocabulary used in it.

### 3.1 Preprocessing

After extracting the mapping between concepts and phrases from the ontology, each phrase is assigned a unique identifier. Then, all occurrences of the mapped phrases in the training corpus are replaced with the unique identifiers, converting them to unigrams. For example, in the sentence

```
Patient was diagnosed with chronic
obstructive lung disease in December.
```

the phrase "chronic obstructive lung disease" will be replaced with its unique identifer, yielding

```
Patient was diagnosed with PHRASE_1337
              in December.
```

This replacement is done greedily over the number of tokens matched, producing two parallel versions of the training corpus: the original, untagged text ($T_U$), and the text tagged for mapped phrases ($T_T$). One token in $T_T$ may correspond to multiple tokens in $T_U$.

## 3.2 Proposed model

Our model adapts the skip-gram with negative sampling variant of `word2vec` for joint embedding of concepts, phrases, and words. Embeddings of dimensionality $d$ utilize three matrices:

- $E_W = |W| \times d$; word embeddings

- $E_P = |P| \times d$; phrase embeddings

- $E_C = |C| \times d$; concept embeddings

Additionally, we use a single negative sampling matrix $E_{NS} = |W| \times k$. The relationships between these matrices are illustrated in Figure 2.

To train the embeddings, we iterate over the tagged and untagged versions of the training corpus in parallel, using a sliding context window of $k$ untagged words. Word embeddings are trained using all words around them, and phrase embeddings are trained using the word contexts around the complete phrase. Concept embeddings are updated using the contexts of phrases that can represent them; for a mention of phrase $p$, these updates are normalized by the number of concepts $p$ can represent. In all cases, the embeddings are trained to maximize the log-likelihood of their observed contexts, and to minimize the log-likelihood of randomly-selected negative samples.

### 3.2.1 Training objective

To calculate the loss for a single $(w, p, o)$ observation, then, where $w$ is the observed word, $p$ is

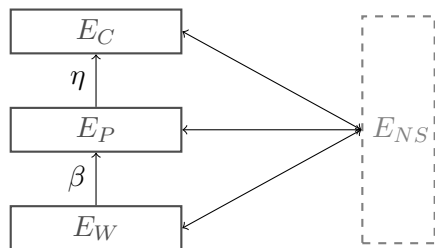

Figure 2: Schematic of proposed model; the dashed $E_{NS}$ box is discarded after training.

the observed phrase just completed, and $o$ is the observed context word, we do as follows. Let $\mathcal{C}_p$ be the set of concepts that $p$ can represent, and $\mathcal{N}$ be the set of negative samples for this observation. We follow the notation of (Levy and Goldberg, 2014) in representing the probability of the pair $(w, o)$ coming from the true data distribution:

$$P(D = 1|w, c) = \sigma(\vec{w} \cdot \vec{c}) = \frac{1}{1 + e^{-\vec{w} \cdot \vec{c}}} \quad (1)$$

Then, the word embedding loss $\ell_w$, phrase embedding loss $\ell_p$, and concept embedding loss $\ell_c$ can be calculated as follows:

$$\ell_w = \log\sigma(\vec{w} \cdot \vec{o}) - \sum_{n \in \mathcal{N}} \log\sigma(\vec{w} \cdot \vec{n}) \quad (2)$$

$$\ell_p = \log\sigma(\vec{p} \cdot \vec{o}) - \sum_{n \in \mathcal{N}} \log\sigma(\vec{p} \cdot \vec{n}) \quad (3)$$

$$\ell_{c \in \mathcal{C}_s} = \left[ \log\sigma(\vec{c} \cdot \vec{o}) - \sum_{n \in \mathcal{N}} \log\sigma(\vec{c} \cdot \vec{n}) \right] \frac{1}{|\mathcal{C}_s|} \quad (4)$$

### 3.2.2 Enforcing embedding similarity

The training algorithm described in Section 3.2 shares information between words, phrases, and concepts only via the shared negative sample embedding matrix. This can be beneficial, as some phrases are not easily decomposable; in the case of "Lou Gehrig's disease", knowing that Lou Gehrig played baseball says little about ALS. However, many phrases have some compositionality (e.g., "high blood pressure"), and many phrases directly encode useful information about the concept (e.g., "amyotrophic lateral sclerosis").

To enable a tradeoff between independence and compositionality in our training, we introduce two hyperparameters to our model: $\beta$ controls the compositionality of phrases by words, and $\eta$ controls the compositionality of concepts by phrases.

To construct the updated loss function, let $\mathcal{W}_p$ be the list of words in phrase $p$, and $\overline{\mathcal{W}_p}$ be the average of their embeddings. For a given concept $c$, let $\mathcal{P}_c$ denote the set of phrases that can represent it, with $\overline{\mathcal{P}_c}$ being the average of their embeddings.

Now, the updated objective functions look like:

$$\ell_w = \log\sigma(\vec{w} \cdot \vec{o}) - \sum_{n \in \mathcal{N}} \log\sigma(\vec{w} \cdot \vec{n}) \quad (5)$$

$$\ell_p = (1 - \beta)\Big(\log\sigma(\vec{p} \cdot \vec{o}) - \sum_{n \in \mathcal{N}} \log\sigma(\vec{p} \cdot \vec{n})\Big)$$
$$+ \beta\big(\log\sigma(\vec{p} \cdot \overline{\mathcal{W}_p})\big) \quad (6)$$

$$\ell_{c \in \mathcal{C}_s} = \Big[(1 - \eta)\big(\log\sigma(\vec{c} \cdot \vec{o}) - \sum_{n \in \mathcal{N}} \log\sigma(\vec{c} \cdot \vec{n})\big)$$
$$+ \eta\big(\log\sigma(\vec{c} \cdot \overline{\mathcal{P}_c})\big)\Big]\frac{1}{|\mathcal{C}_s|} \quad (7)$$

## 4 Materials

### 4.1 Training corpora

We train our embeddings on four corpora. Prior work has noted the effects of training corpus choice on a variety of tasks in the biomedical domain (Pakhomov et al., 2016; Griffis et al., 2016; Garla and Brandt, 2012), so we experiment with literature abstracts, clinical narratives, and their combination. Table 1 lists statistics about concepts and phrases embedded from each corpus.

PUBMED All abstracts from the 2016 PubMed baseline of biomedical literaure (i.e., all abstracts present in PubMed at the beginning of 2016).

CLINICAL A set of clinical notes from the Ohio State Wexner Medical Center, regarding patients diagnosed with a variety of chronic conditions.

PM+CL Concatenation of the above corpora.

WIKINYT Combination of all articles in a 2015 dump of English Wikipedia and the New York Times portions of Gigaword (Parker et al., 2011).

### 4.2 Ontologies

Our canonical concepts and their representative strings are sourced from two ontologies: the Unified Medical Language System (UMLS) (Bodenreider, 2004), and YAGO (Suchanek et al., 2007; Mahdisoltani et al., 2015). The UMLS Metathesaurus provides mappings between more than 100 biomedical vocabularies, enumerates the medical concepts described, and includes canonical forms (Concept Unique Identifiers, or CUIs) for each concept. CUIs are further mapped to one or more textual forms, along with semantic type information. YAGO is a knowledge base encoding information about real-world entities and the relationships between them, including information from

WordNet (Fellbaum, 1998) and Wikipedia. As with the UMLS, YAGO maps between canonical entity identifiers and one or more textual forms.

In both cases, we restrict ourselves to text forms in English, and entities with at least one text form that appears $\geq 25$ times in our corpora.

### 4.3 Evaluation datasets

#### 4.3.1 Biomedical similarity and relatedness

To evaluate our biomedical concepts, we used the MayoSRS and Mini Mayo SRS datasets from (Pedersen et al., 2007) and the UMNSRS Similarity and Relatedness datasets of (Pakhomov et al., 2010). MayoSRS consists of 101 pairs of concepts, identified by both UMLS CUI and text form, ranked on a scale of 1-4; the Mini Mayo SRS set includes 29 of these pairs, selected for high inter-rater agreement. The UMNSRS Similarity and Relatedness datasets are composed of 566 and 587 concept pairs, respectively, ranked on a scale of 1-1600 for similarity or relatedness; CUI and text forms are provided for all concepts.

#### 4.3.2 General-domain similarity and relatedness

While there are well-studied datasets of word similarity and relatedness available, such as WordSim-353 (Finkelstein et al., 2001), SimLex-999 (Hill et al., 2015), as well as datasets for evaluating compositionality (Marelli et al., 2014) and typed relations between nouns (Baroni and Lenci, 2011), we are not aware of comparable similarity or relatedness datasets for real-world entities. We therefore present two novel datasets composed of human judgments of similarity and relatedness between pairs of people, places, and organizations.

We used Amazon's Mechanical Turk crowdsourcing platform for our data collection. Participants were asked to assign a similarity or relatedness score from 0-100 to each of 34 pairs of entities, with a higher score indicating higher similarity/relatedness. We collected judgments for 24 such sets of entity pairs, getting separate judgments for similarity and for relatedness. 4 entity pairs in each set were used as validation questions, as they were determined to tend strongly towards either high or low similarity/relatedness. These pairs were not included in the final dataset; filtering for participants who self-reported high English reading proficiency and who gave reasonable responses to the validation questions pro-

| Corpus | Onto | # Docs | # Tokens | # Concepts | Max Phrases | Avg Phrases |
|---|---|---|---|---|---|---|
| PUBMED | | 14.7m | 2.7b | 188,486 | 390 | 5.4 |
| CLINICAL | UMLS | 377k | 160.3m | 37,219 | 160 | 7.2 |
| PM+CL | | 15.1m | 2.8b | 193,064 | 390 | 5.4 |
| WIKINYT | YAGO | 6.2m | 3.5b | 748,752 | 7,191 | 3.2 |

Table 1: Number of documents and tokens in each of the corpora used for training concept embeddings, along with the number of concepts (from ontology Onto) found in each. Max Phrases is the maximum number of representative phrases for a single concept that appear at least 25 times in the corpus; Avg Phrases is the average number of mapped phrases over all concepts.

duced a final dataset of 688 pairs for each task (658 pairs present in both filtered datasets). Participants were paid at least $0.75 for each survey, with the rate increased as needed to guarantee state minimum wage for the time spent on the survey. Overall Fleiss' $\kappa$ for our annotations was 0.24 for both datasets, indicating fair agreement between annotators. For further details about our data collection and analysis of response, please see the supplemental material.

## 5 Experiments

### 5.1 Model comparisons

To evaluate the benefit of embedding concepts directly, we compare with several approximations using embeddings learned on our corpora:

- APPROXPHR$_{co}$ We approximate a concept's embedding as the average of the embeddings of its representative strings.

- APPROXWORD We approximate a concept's embedding as the average embedding of the words used in it representative strings.

- PHRASE$_{co}$ We use the learned embedding of a particular string directly.

- WORD We approximate a particular string as the average embedding of its words.

| Domain | Dataset | # of pairs |
|---|---|---|
| Medical | Mayo SRS | 101 |
| | Mini Mayo SRS | 29 |
| | UMNSRS Relatedness | 587 |
| | UMNSRS Similarity | 566 |
| General | Relatedness | 688 |
| | Similarity | 688 |

Table 2: Similarity and relatedness datasets used

In all settings, we use subscript $_{co}$ to denote jointly-trained embeddings; WORD$_{ind}$ indicates word embeddings trained alone using the skip-gram model of `word2vec`.

In addition, we compare with a variety of pre-trained embeddings from prior research:

- DEVINE Embeddings of 52k UMLS CUIs trained over sequences of CUIs automatically tagged from EHR data and medical abstracts (De Vine et al., 2014).

- CHOI Embeddings of 15k UMLS CUIs trained via hierarchical sampling over sequences of manually annotated claims codes from EHR data (Choi et al., 2016b).

- MENCIA Embeddings of 26k MESH headings trained over manually-tagged PubMed abstracts to maximize similarity of headings to relevant documents (Mencia et al., 2016).

- CHIU-2 Word embeddings trained on PubMed abstracts with a context window size of 2 (Chiu et al., 2016a).

- CHIU-30 Word embeddings trained on PubMed abstracts with a context window size of 30 (Chiu et al., 2016a).

### 5.2 Hyperparameter tuning

Recent findings highlight the importance of hyperparameter selection for performance of distributed representations on downstream tasks (Levy et al., 2015; Chiu et al., 2016a). We therefore evaluate the impact of various hyperparameters on the similarity and relatedness task, and use the results to tune optimal settings for further evaluations.[2] In

---

[2](Chiu et al., 2016a) point out that in the biomedical domain, hyperparameter settings that improve performance on intrinsic evaluations of the vector space may degrade perfor-

| Corpus | $\beta$ | $\eta$ | NS | WS | Dim |
|---|---|---|---|---|---|
| PUBMED | 0.75 | 0.0 | 5 | 30 | 400 |
| CLINICAL | 0.25 | 0.25 | 10 | 5 | 600 |
| PM+CL | 0.0 | 0.0 | 5 | 30 | 100 |
| WIKINYT | 0.0 | 0.0 | 5 | 5 | 300 |

Table 3: Tuned hyperparameters for each corpus; NS is the number of negative samples used, WS is the size of the context window, and Dim is the dimensionality of the vector space.

addition to the $\beta$ and $\eta$ tradeoff parameters of our model, we experiment with number of negative samples, vector dimensionality, and context window size. In all cases, we trained for 10 iterations. Table 3 lists the final tuned settings for each corpus; for more details on our hyperparameter tuning, please see the supplementary material.

### 5.3 Similarity and relatedness

Given a pair of ontology entities $\langle e_1, e_2 \rangle$, we calculate both their similarity and relatedness via the cosine similarity of their embeddings. We then rank each list of entity pairs in order of decreasing cosine similarity, and compare our ranking against the ranked list of human similarity or relatedness judgments. We report Spearman's rank correlation coefficient (Spearman's $\rho$), which ranges from -1 (reverse ranking) to 1 (same ranking); Table 4 shows our results. On the two smaller biomedical datasets, our method is outperformed by CHOI; however, on the larger datasets, our concepts embeddings have the highest correlation with human judgments on the similarity task, and competitive performance on the relatedness task (jointly embedded phrases are negligibly better than CHIU-2). On our real-world entity datasets, the phrase-based concept approximation outperforms the direct concept embeddings.

### 5.4 Concepts, phrases, and words

Since our model jointly embeds concepts, phrases, and words into the same space, we assess how well the ontological links between concepts and phrases and between phrases and words are preserved. We first approximate each concept and phrase by averaging the embeddings of their representative phrases and words, and calculate co-

mance in extrinsic downstream applications. Our analysis is restricted to evaluations relying on the affine organization of the vector space; thus, we note that these optimized hyperparameters will likely change for downstream tasks.

sine similarity with the learned concept or phrase embedding. Figure 3a shows that almost every concept has very high similarity to its mean phrasal embedding, but we see higher variance when comparing phrases to the average of their words (Figure 3b). This is further reflected in Figure 3c, where we see that most concepts has cosine similarity near 0.5 to the average of the words of its representative phrases.

To assess if this pattern holds for individual cases, we examine the phrases and words in the neighborhood of a parent concept or phrase. In particular, given a parent concept or phrase $p$, we calculate the mean average precision (MAP) of $C_p$, the child phrases or words connected to it, within the full phrase or word vocabulary $V$.

We first rank all $v \in V$ in order of descending cosine similarity to $p$; this produces the ranked list $\hat{V}$. Then, for each child term $c$ in the "correct" subset $C_p \subset V$, we calculate its precision as:

$$Pr(c|C_p, V) = \frac{1_{\hat{V}_{<c}}}{\hat{V}_c} \qquad (8)$$

where $\hat{V}_c$ denotes the index of $c$ in $\hat{V}$, and $1_{\hat{V}_{<c}}$ is an indicator function over the elements of $\hat{V}$ up to $c$ that is 1 if the element is in $C_p$ and 0 otherwise. The MAP calculation is then

$$\text{MAP}(p|V) = \frac{1}{|P|} \sum_{p \in P} \frac{\sum_{c \in C_p} Pr(c|C_p, V)}{|C_p|} \qquad (9)$$

Figure 4a shows that the resulting MAP scores for concepts and their phrases are skewed towards 0, with clear peaks at 0.5 and 1. This suggests that most concepts are not especially near their individual representative phrases in the vector space, in comparison to the full phrase vocabulary. In contrast, the MAP scores for phrases and their words are strikingly bi-modal near 0 and 1, indicating that most phrases are either quite close or very distant from their component words in the embedding space. Taken together, these results indicate that our model seems to be distributing component phrases or words fairly equally around their parents, though at varying distance.

### 5.5 Semantic type clustering

Choi et al. (2016b) propose the Medical Conceptual Similarity Measure, which evaluates how well embedded concepts of the same semantic

| Corpus | Setting | Mayo SRS | Mini Mayo SRS | UMNSRS Relatedness | UMNSRS Similarity |
|---|---|---|---|---|---|
| PM+CL | $\text{WORD}_{ind}$ | 0.237 | 0.400 | 0.546 | 0.567 |
| | $\text{WORD}_{co}$ | 0.370 | 0.505 | 0.579 | 0.615 |
| | $\text{PHRASE}_{co}$ | 0.618 | 0.477 | **0.591** | 0.628 |
| | $\text{APPROXWORD}_{ind}$ | 0.376 | 0.606 | 0.375 | 0.445 |
| | $\text{APPROXWORD}_{co}$ | 0.493 | 0.600 | 0.400 | 0.495 |
| | $\text{APPROXPHR}_{co}$ | 0.609 | 0.623 | 0.543 | 0.637 |
| | $\text{CONCEPT}_{co}$ | 0.593 | 0.591 | 0.559 | **0.653** |
| Concept baselines | DEVINE(CUIs only) | 0.559 | 0.434 | 0.422 | 0.455 |
| | CHOI(+ICD9,etc.) | **0.817** | **0.726** | 0.384 | 0.552 |
| | MENCIA(MeSH headers) | 0.646 | 0.572 | 0.534 | 0.565 |
| Word baselines | CHIU-2(words only) | 0.368 | 0.565 | 0.496 | 0.560 |
| | CHIU-30(words only) | 0.472 | 0.630 | 0.590 | 0.653 |

Table 4: Similarity/relatedness results for tuned embeddings on biomedical datasets. Reported scores are Spearman's $\rho$, which ranges $[-1, 1]$. We only report PM+CL results here; PUBMED and CLINICAL are similar. Best-performing settings for each dataset are marked in bold.

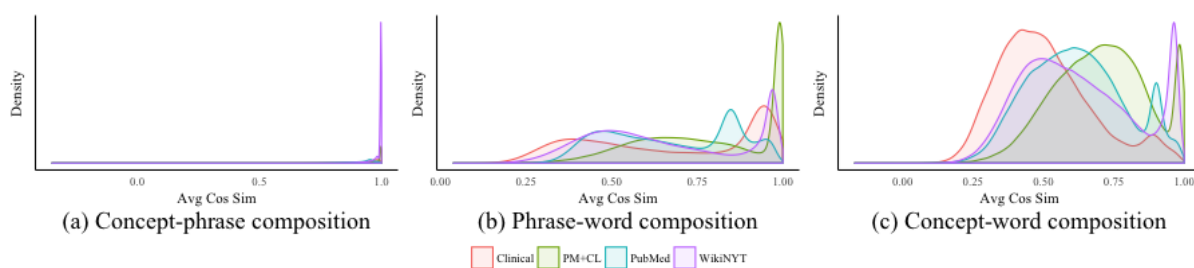

(a) Concept-phrase composition (b) Phrase-word composition (c) Concept-word composition

Clinical PM+CL PubMed WikiNYT

Figure 3: Compositionality evaluation

| Setting | Relatedness | Similarity |
|---|---|---|
| $\text{WORD}_{ind}$ | 0.602 | 0.589 |
| $\text{WORD}_{co}$ | 0.645 | 0.636 |
| $\text{PHRASE}_{co}$ | 0.784 | 0.743 |
| $\text{APPROXWORD}_{ind}$ | 0.578 | 0.588 |
| $\text{APPROXWORD}_{co}$ | 0.615 | 0.624 |
| $\text{APPROXPHR}_{co}$ | **0.810** | **0.766** |
| $\text{CONCEPT}_{co}$ | 0.789 | 0.750 |

Table 5: Similarity/relatedness results for tuned embeddings on real-world entity datasets.

type cluster in a fixed neighborhood of $k$ nearest neighbors. To obtain a more complete picture of the space, we adapt their methodology to find all concepts of a given semantic type within the entire space; as this is a ranking problem, we report mean average precision. We follow Choi et al. (2016b) in selecting 6 common semantic types to evaluate: disease or syndrome, pharmacologic substance, neoplastic process, finding, injury and poisoning, and sign or symptom (this replaces clinical drug, as not all concept embedding sets contained multiple concepts of that type).

We evaluated our three biomedical concept embeddings, and compared against the four medical concept embeddings from prior work; the results are shown in Table 6. The medical code embeddings of Choi et al. (2016b) demonstrate superior clustering in aggregate,[3] but clustering performance is mixed among the semantic types, with our embeddings clustering competitively with prior work. In particular, "sign or symptom," which has many exemplars in each embedding set, is uniformly poorly clustered, suggesting that its semantics are difficult to capture.

## 6 Discussion

Despite the noisiness of our distantly-supervised training method, our results indicate that we capture certain semantic characteristics of ontology concepts nearly as well as prior methods that require significantly more annotation. Furthermore, we can scale to a much larger vocabulary without additional expense. However, there are a number of points that bear further examination.

Our results comparing $\text{APPROXWORD}_{co}$ to $\text{APPROXWORD}_{ind}$ align with prior work indicating that including structured knowledge about concepts in training word embeddings improves

---

[3]Choi et al. (2016b) propose another embedding technique that has superior MAP (0.20) on finding, which we omit for brevity.

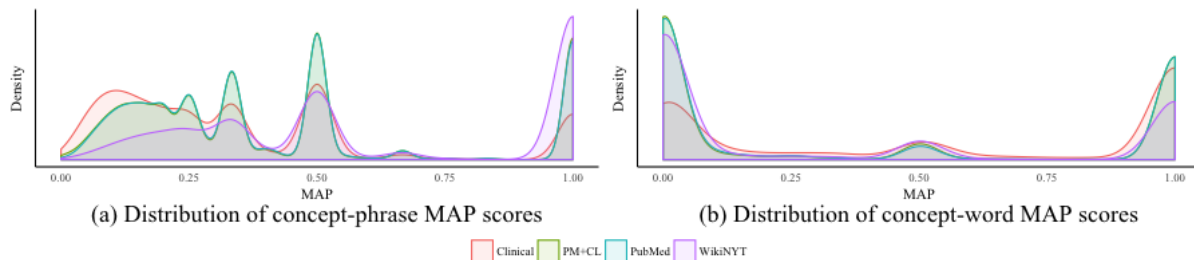

Figure 4: Mean average precision distributions for concept-phrase and phrase-word comparisons.

| STY | PM | CL | PM +CL | DV | Choi | ME |
|---|---|---|---|---|---|---|
| pharmacologic substance | **0.26** | 0.14 | 0.20 | 0.17 | 0.21 | 0.13 |
| disease or syndrome | 0.20 | 0.12 | 0.15 | 0.21 | **0.33** | 0.16 |
| injury or poisoning | 0.04 | 0.02 | 0.03 | 0.04 | **0.36** | 0.05 |
| sign or symptom | **0.09** | 0.06 | 0.05 | 0.06 | 0.02 | 0.02 |
| neoplastic process | 0.20 | 0.04 | 0.18 | 0.14 | **0.25** | 0.10 |
| finding | 0.12 | 0.12 | 0.11 | 0.09 | **0.12** | 0.01 |
| Overall | 0.15 | 0.08 | 0.12 | 0.12 | **0.22** | 0.08 |

Table 6: MAP for semantic type clustering with selected biomedical semantic types (STY). PM=PUBMED, CL=CLINICAL, DV=DeVine, ME=Mencia. Highest MAP per semantic type is marked in bold.

their performance in some semantic tasks (Wang et al., 2014a; Faruqui et al., 2015). However, the extreme similarity observed in our compositionality analysis, and the high performance of concept approximations, suggests that some non-lexical information about concepts is not being captured in our method. Leveraging an approach similar to Wang et al. (2014a) and directly including ontology structure as a training criterion may be one way to model some of this additional information.

Chiu et al. (2016b,a); Faruqui et al. (2016) describe several issues with evaluating linguistic embeddings with similarity and relatedness tasks, and poor generalization of results to downstream applications. While we address some of their concerns (e.g., our concept embeddings rule out polysemy), several of the issues they describe remain. One of these is the lack of train/test splits on the similarity and relatedness tasks; though we do not train our model directly on these data, we are still tuning on the test set. We attempt to mitigate this issue by an adapted 5-fold cross-validation, and our other analyses of the space indicate that our embeddings are capturing some semantic properties, but applications to other intrinsic tasks such as analogy completion or downstream NLP tasks would provide additional strong evidence of these properties. While analogies at the entity/concept level are limited, we suggest that applications such as entity linking be pursued in future work.

Finally, we note a slight mismatch between the ontologies we used. The UMLS encodes information about abstract biomedical concepts, while YAGO is focused on concrete, real-world entities. We demonstrate that our method is able to model both kinds of information, but it would benefit from an application to more abstract data in the general domain, similar to previous work with BabelNet (Camacho-Collados et al., 2015).

## 7  Conclusions

We propose a novel model to jointly embed canonical concepts, the phrases that represent them, and words into a shared vector space. Our method only uses distant supervision from phrases linked to concepts in an ontology, but we experimentally demonstrate that our embeddings preserve several semantic properties comparably to recent methods that require human-annotated data. In particular, our concept embeddings maintain similarity and relatedness, as evaluated by cosine similarity, and preserve the links between concepts of the same type and between concepts and their representative phrases.

With this paper, we include `cui2vec`, a software implementation of our method, as well as our novel dataset of similarity and relatedness of real-world entities.

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

## A Supplemental Material

### A.1 Hyperparameter tuning

We used an adapted 5-fold cross-validation for our hyperparameter tuning. First, each dataset was partitioned into 5 portions. Then, each hyperparameter setting was evaluated on each of the other 4 partitions, and the average of those results was considered to be that setting's error for the target partition. The overall error of each setting is then the average of its 5 per-fold errors.[4]

We evaluated the following settings: $\beta, \eta \in \{0, 0.25, 0.5, 0.75, 1\}$, number of negative samples $\in \{5, 10, 15\}$, vector dimension $\in \{50, 100, 200, 300, 400, 500, 600\}$, and context window size $\in \{2, 5, 10, 30\}$. Default hyperparameter settings were 5 negative samples, window size of 2, and vector dimension of 100. Chiu et al. (2016a) found that a learning rate of $\alpha = 0.05$ performed better than the standard 0.025 with biomedical tasks; accordingly, we use $\alpha = 0.05$ in our biomedical corpora. However, the traditional learning rate of $\alpha = 0.025$ gave better performance on our general-domain corpus. In all cases, we restricted ourselves to a minimum frequency threshold for words and phrases of 25[5].

In general, we find that while enforcing some small amount of concept-phrase or phrase-word with the $\beta$ and $\eta$ parameters can be helpful, similarity and relatedness performance trend negative as $\beta$ and $\eta$ increase. This negative trend is more pronounced with the concept-phrase similarity, controlled by the $\eta$ parameter; a non-zero $\eta$ is only optimal in a single corpus setting.

With regards to the other hyperparameters we evaluated, no clear trends emerged, as each hyperparameter behaved differently in each corpus. However, the differences between best and worst-performing hyperparameter settings were relatively small (the largest was a difference in Spearman's $\rho$ of .075, from vector dimensionality in

---

[4]Since the similarity and relatedness evaluation is a ranking task, evaluated by Spearman's $\rho$, this is not equivalent to simply ranking the full dataset; in fact, the validation numbers skew worse than the overall numbers, as the smaller dataset size in the cross-fold evaluations penalizes ranking errors more highly.

[5]This was done primarily for memory constraints. However, given the small size of our Clinical corpus, we experimented with smaller thresholds; we found overall decreases in performance and only small gains in the number of concepts modeled.

the CLINICAL corpus). The most consistently-performing hyperparameter settings across the corpora were negative samples=5, vector dimensionality=100, and context window size=5.

## A.2 Similarity/Relatedness dataset

We followed a similar process to Pakhomov et al. (2010) in selecting the entity pairs to be used in our dataset. We first filtered all entities in YAGO to the subset that we learned embeddings for ($\approx$ 700k); we then filtered to only entities labeled with WordNet types organization or person, or with the YAGO type geoEntity. For each pairing of these categories (Organization-Organization, Organization-Place, Organization-Person, Place-Place, Place-Person, and Person-Person), we manually selected 30 pairs of entities for each of the following relatedness categories: Completely Unrelated, Somewhat Unrelated, Somewhat Related, and Highly Related. These produced the list of 720 entity pairs we used for our Mechanical Turk surveys.

We augmented each survey of 30 questions with 4 manually-created validation pairs using common entities (e.g., London, New York), each of which was categorized as Highly Related or Completely Unrelated. We included these validation questions at random indices in our surveys. To evaluate if participants were reading the questions, we binned their ratings on these validation questions into 0-25 (Completely Unrelated), 26-50 (Somewhat Unrelated), 51-75 (Somewhat Related), and 76-100 (Highly Related). If a participant's ratings disagreed with ours on multiple validation questions, we discarded their data (we allowed disagreement on a single question, as some validation questions had high variance in responses among reliable annotators).

We recruited 6 participants for each survey, for a total of 34 unique participants across the 48 HITs. The majority responded to a single HIT, while 3 completed more than 20. We discarded all submissions from 3 participants, as they did not report English reading proficiency (1) or did not satisfy the validation questions (2). All participants were paid, regardless of if we used their data or not.

To generate the final dataset, we assessed each participant's responses to the validation questions in each survey. We kept surveys for which we had at least 4 participants with satisfactory answers to

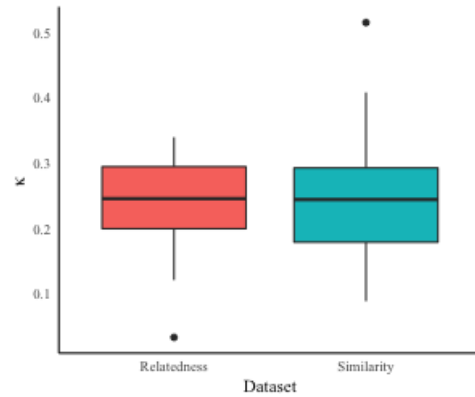

Figure 5: Distribution of Fleiss' $kappa$ for the HITs in our similarity and relatedness datasets.

the validation questions; this resulted in discarding 1 of the 24 HITs for each task. Due to 2 repeated pairs, this gave us final dataset sizes of 688 pairs for each of similarity and relatedness, 658 of which were shared between the tasks.

In assessing inter-annotator agreement (IAA), we considered each HIT individually, as we had neither the same participants nor the same number of accepted responses for all HITs. Figure 5 shows the distribution of Fleiss' $\kappa$ for the HITs in each dataset; the average for each case was 0.24.

