# Peer review of "A Weakly-Supervised Method for Jointly Embedding Concepts, Phrases, and Words"

_ACL 2017 — decision unknown_

[Official Review · Reviewer 1 · rating 2 · confidence 3]
soundness 4 · originality 3 · clarity 4 · impact 3 · substance 3 · appropriateness 4 · meaningful comparison 3 · presentation format Poster

The paper describes an extension of word embedding methods to also provide
representations for phrases and concepts that correspond to words.  The method
works by fixing an identifier for groups of phrases, words and the concept that
all denote this concept, replace the occurrences of the phrases and words by
this identifier in the training corpus, creating a "tagged" corpus, and then
appending the tagged corpus to the original corpus for training.  The
concept/phrase/word sets are taken from an ontology.  Since the domain of
application is biomedical, the related corpora and ontologies are used.  The
researchers also report on the generation of a new test dataset for word
similarity and relatedness for real-world entities, which is novel.

In general, the paper is nicely written.  The technique is pretty natural,
though not a very substantial contribution. The scope of the contribution is
limited, because of focused evaluation within the biomedical domain.

More discussion of the generated test resource could be useful.  The resource
could be the true interesting contribution of the paper.

There is one
small technical problem, but that is probably just a matter of mathematical
expression than implementation.

Technical problem:
Eq. 8: The authors want to define the MAP calculation.                          This
is a
good
idea,
thought I think that a natural cut-off could be defined, rather than ranking
the entire vocabulary.                          Equation 8 does not define a
probability;
it is
quite
easy to show this, even if the size of the vocabulary is infinite.  So you need
to change the explanation (take out talk of a probability).

Small corrections:
line:
556: most concepts has --> most concepts have

[Official Review · Reviewer 2 · rating 2 · confidence 4]
soundness 3 · originality 2 · clarity 4 · impact 3 · substance 4 · appropriateness 5 · meaningful comparison 3 · presentation format Poster

Summary: This paper presents a model for embedding words, phrases and concepts
into vector spaces. To do so, it uses an ontology of concepts, each of which is
mapped to phrases. These phrases are found in text corpora and treated as
atomic symbols. Using this, the paper uses what is essentially the skip-gram
method to train embeddings for words, the now atomic phrases and also the
concepts associated with them. The proposed work is evaluated on the task of
concept similarity and relatedness using UMLS and Yago to act as the backing
ontologies.

Strengths:

The key question addressed by the paper is that phrases that are not lexically
similar can be semantically close and, furthermore, not all phrases are
compositional in nature. To this end, the paper proposes a plausible model to
train phrase embeddings. The trained embeddings are shown to be competitive or
better at identifying similarity between concepts.

The software released with the paper could be useful for biomedical NLP
researchers.

- Weaknesses:

The primary weakness of the paper is that the model is not too novel. It is
essentially a tweak to skip-gram. 

Furthermore, the full model presented by the paper doesn't seem to be the best
one in the results (in Table 4). On the two Mayo datasets, the Choi baseline is
substantially better. A similar trend seems to dominate Table 6 too. On the
larger UMNSRS data, the proposed model is at best competitive with previous
simpler models (Chiu).

- General Discussion:

The paper says that it is uses known phrases as distant supervision to train
embeddings. However, it is not clear what the "supervision" here is. If I
understand the paper correctly, every occurrence of a phrase associated with a
concept provides the context to train word embeddings. But this is not
supervision in the traditional sense (say for identifying the concept in the
text or other such predictive tasks). So the terminology is a bit confusing.

 The notation introduced in Section 3.2 (E_W, etc) is never used in the rest of
the paper.

The use of \beta to control for compositionality of phrases by words is quite
surprising. Essentially, this is equivalent to saying that there is a single
global constant that decides "how compositional" any phrase should be. The
surprising part here is that the actual values of \beta chosen by cross
validation from Table 3 are odd. For PM+CL and WikiNYT, it is zero, which
basically argues against compositionality. 

The experimental setup for table 4 needs some explanation. The paper says that
the data labels similarity/relatedness of concepts (or entities). However, if
the concepts-phrases mapping is really many-to-many, then how are the
phrase/word vectors used to compute the similarities? It seems that we can only
use the concept vectors.

In table 5, the approximate phr method (which approximate concepts with the
average of the phrases in them) is best performing. So it is not clear why we
need the concept ontology. Instead, we could have just started with a seed set
of phrases to get the same results.

[Official Review · Reviewer 3 · rating 2 · confidence 4]
soundness 3 · originality 3 · clarity 3 · impact 3 · substance 4 · appropriateness 5 · meaningful comparison 3 · presentation format Poster

The authors presents a method to jointly embed words, phrases and concepts,
based on plain text corpora and a manually-constructed ontology, in which
concepts are represented by one or more phrases. They apply their method in the
medical domain using the UMLS ontology, and in the general domain using the
YAGO ontology. To evaluate their approach, the authors compare it to simpler
baselines and prior work, mostly on intrinsic similarity and relatedness
benchmarks. They use existing benchmarks in the medical domain, and use
mechanical turkers to generate a new general-domain concept similarity and
relatedness dataset, which they also intend to release. They report results
that are comparable to prior work.

Strengths:

- The proposed joint embedding model is straightforward and makes reasonable
sense to me. Its main value in my mind is in reaching a (configurable) middle
ground between treating phrases as atomic units on one hand to considering
their
compositionallity on the other. The same approach is applied to concepts being
‘composed’ of several representative phrases.

-  The paper describes a decent volume of work, including model development,
an additional contribution in the form of a new evaluation dataset, and several
evaluations and analyses performed.

Weaknesses:

- The evaluation reported in this paper includes only intrinsic tasks, mainly
on similarity/relatedness datasets. As the authors note, such evaluations are
known to have very limited power in predicting the utility of embeddings in
extrinsic tasks. Accordingly, it has become recently much more common to
include at least one or two extrinsic tasks as part of the evaluation of
embedding models.

- The similarity/relatedness evaluation datasets used in the paper are
presented as datasets recording human judgements of similarity between
concepts. However, if I understand correctly, the actual judgements were made
based on presenting phrases to the human annotators, and therefore they should
be considered as phrase similarity datasets, and analyzed as such.

- The medical concept evaluation dataset, ‘mini MayoSRS’ is extremely small
(29 pairs), and its larger superset ‘MayoSRS’ is only a little larger (101
pairs) and was reported to have a relatively low human annotator agreement. The
other medical concept evaluation dataset, ‘UMNSRS’, is more reasonable in
size, but is based only on concepts that can be represented as single words,
and were represented as such to the human annotators. This should be mentioned
in the paper and makes the relevance of this dataset questionable with respect
to representations of phrases and general concepts. 

- As the authors themselves note, they (quite extensively) fine tune their
hyperparameters on the very same datasets for which they report their results
and compare them with prior work. This makes all the reported results and
analyses questionable.

- The authors suggest that their method is superb to prior work, as it achieved
comparable results while prior work required much more manual annotation. I
don't think this argument is very strong because the authors also use large
manually-constructed ontologies, and also because the manually annotated
dataset used in prior work comes from existing clinical records that did not
require dedicated annotations.

- In general, I was missing more useful insights into what is going on behind
the reported numbers. The authors try to treat the relation between a phrase
and its component words on one hand, and a concept and its alternative phrases
on the other, as similar types of a compositional relation. However, they
are different in nature and in my mind each deserves a dedicated analysis. For
example, around line 588, I would expect an NLP analysis specific to the
relation between phrases and their component words. Perhaps the reason for the
reported behavior is dominant phrase headwords, etc. Another aspect that was
absent but could strengthen the work, is an investigation of the effect of the
hyperparameters that control the tradeoff between the atomic and compositional
views of phrases and concepts.

General Discussion:

Due to the above mentioned weaknesses, I recommend to reject this submission. I
encourage the authors to consider improving their evaluation datasets and
methodology before re-submitting this paper.

Minor comments:

- Line 069: contexts -> concepts

- Line 202: how are phrase overlaps handled?

- Line 220: I believe the dimensions should be |W| x d. Also, the terminology
‘negative sampling matrix’ is confusing as the model uses these embeddings
to represent contexts in positive instances as well.

- Line 250: regarding ‘the observed phrase just completed’, it not clear to
me how words are trained in the joint model. The text may imply that only the
last words of a phrase are considered as target words, but that doesn’t make
sense. 

- Notation in Equation 1 is confusing (using c instead of o)

- Line 361: Pedersen et al 2007 is missing in the reference section.

- Line 388: I find it odd to use such a fine-grained similarity scale (1-100) 
for human annotations.

- Line 430: The newly introduced term ‘strings’ here is confusing. I
suggest to keep using ‘phrases’ instead.

- Line 496: Which task exactly was used for the hyper-parameter tuning?
That’s important. I couldn’t find that even in the appendix.

- Table 3: It’s hard to see trends here, for instance PM+CL behaves rather
differently than either PM or CL alone. It would be interesting to see
development set trends with respect to these hyper-parameters.

- Line 535: missing reference to Table 5.